# Two-Dimensional Heterostructure of PPy/CNT–*E. coli* for High-Performance Supercapacitor Electrodes

**DOI:** 10.3390/ma15175804

**Published:** 2022-08-23

**Authors:** Kwang Se Lee, Jung Yong Kim, Jongwook Park, Jang Myoun Ko, Sharon Mugobera

**Affiliations:** 1Department of Advanced Materials & Chemical Engineering, Kyungnam College of Information & Technology, 45 Jurye-ro, Busan 47011, Sasang-gu, Korea; 2Department of Materials Science and Engineering, Adama Science and Technology University, P.O.Box 1888, Adama, Ethiopia; 3Center of Advanced Materials Science and Engineering, Adama Science and Technology University, P.O.Box 1888, Adama, Ethiopia; 4Integrated Engineering, Department of Chemical Engineering, Kyung Hee University, Yongin 17104, Gyeonggi, Korea; 5Department of Chemical and Biological Engineering, Hanbat National University, 125 Dongseo-daero, Daejeon 34158, Yuseong-gu, Korea

**Keywords:** polypyrrole, carbon nanotube, bacteria

## Abstract

The nano-biocomposite electrodes composed of carbon nanotube (CNT), polypyrrole (PPy), and *E. coli*-bacteria were investigated for electrochemical supercapacitors. For this purpose, PPy/CNT–*E. coli* was successfully synthesized through oxidative polymerization. The PPy/CNT–*E. coli* electrode exhibited a high specific capacitance of 173 F∙g^−1^ at the current density of 0.2 A∙g^−1^, which is much higher than that (37 F∙g^−1^) of CNT. Furthermore, it displayed sufficient stability after 1000 charge/discharge cycles. The CNT, PPy/CNT, and PPy/CNT–*E. coli* composites were characterized by x-ray diffraction, scanning electron microscopy, and surface analyzer (Brunauer–Emmett–Teller, BET). In particular, the pyrrole monomers were easily adsorbed and polymerized on the surface of CNT materials, as well as *E. coli* bacteria enhanced the surface area and porous structure of the PPy/CNT–*E. coli* composite electrode resulting in high performance of devices.

## 1. Introduction

The electrochemical capacitor (ECC) is an energy storage device, which is also called supercapacitor or ultracapacitor [1,2]. The energy storage devices with a high power and energy density are in high demand for versatile applications, such as industrial equipment, modern electric vehicles, and other small devices [3,4,5]. Basically, supercapacitor can be divided by two types. One is an electric double layer capacitor (EDLC) and the other is a pseudocapacitor. Here, the difference between EDLC and pseudocapacitors is that the former stores energy at the interface between electrolyte and electrode, whereas the latter utilizes additional faradaic reactions at the surface of electrodes in addition to the intrinsic EDLC capacitance [6,7,8].

Numerous studies have focused on creating the most effective electronic gadgets in the smallest possible sizes as a result of the science and technology sector’s ongoing advancement. The interdigital structured micro-supercapacitor is one example. Due to its excellent power density, this tiny supercapacitor is gradually replacing micro-batteries [9]. Micro-batteries have a limited lifespan and must be replaced frequently, which is a hassle when doing an analysis. Due to their superior charge and discharge rates, as well as their longer life term, micro-supercapacitors are mostly used as energy storage devices in the biomedical industry. Paper supercapacitors are another type of supercapacitor that has been of interest lately. They are thin and flexible since they are constructed from paper-based substrates. These characteristics enable their use on walls or billboards without adding bulk to the space. A high operational voltage can be obtained by connecting them in series [10]. Due to their high voltage window, asymmetric supercapacitors have also attracted attention [11,12], EDLC and pseudo-capacitive cathode types in particular. Metal oxides or conducting polymers like polyaniline or polypyrrole can be used as cathode materials. The EDLC//battery-type hybrid supercapacitor is another type [13], which uses metal oxides and hydroxides as the cathode materials and carbon materials as an anode.

As an electrode material for supercapacitors, semiconducting and metallic polymers (i.e., -conjugated materials) have been employed. Among these, polypyrrole (PPy) has become the industry standard material for energy storage due to its numerous benefits, including eco-friendliness, low cost, simple handling, and high capacity [14,15,16,17,18]. Its pyrrole monomer does, however, have significant downsides, such as a low conductivity of 10^−11^–10^−10^ S/cm. Prior to doping with halogens like iodine, bromine, and chlorine, PPy functions as an insulator in its natural condition. The conductivity range of un-doped PPy is 10^−6^ to 10^−3^ S/cm [19,20,21]. Its mechanical performance has various flaws, according to Du et al., and this has a detrimental impact on its electrochemical performance since capacitance degrades [19]. To make up for these drawbacks, carbon-based compounds like graphene and carbon nanotubes (CNT) were developed as structural fillers. Algae, fungi, viruses, and prokaryotic bacteria are examples of adaptable biomaterials that have been employed as a bio-template to enhance the performance of composite materials [11]. Because they are among these microorganisms that are not only plentiful but also mass-producible and renewable, bacteria, in particular, are a focus of attention. As a result, these bacteria have been employed as a model for composite electrodes, providing thin sheets with a clearly defined shape. The interlocking pore networks, heteroatom moiety, and a sizable surface area in composite electrodes may all be directly related to the structure of bacteria [12].

In earlier investigations, Catauro et al. used a sol-gel synthesis process to create antimicrobial biomedical implants. This was accomplished by first embedding various ratios of poly(e-caprolactone) and polyethylene glycol in the silica matrix. Then, substantial quantities of chlorogenic acid were added to each SiO_2_/PEG and SiO_2_/PCL solution. Cell adhesion and growth were boosted by polyethylene glycol polymer. When *E. coli* bacteria were tested for resistance, the materials’ antibacterial activity responded favorably [22].

The electrochemical behavior of carbon nanohorns/polyaniline (PANI)/PPy composites was examined by Chang et al. [13,23], who found that they had outstanding cycling stability after 5000 cycles and a specific capacitance of 762 F∙g^−1^ at 5 m∙Vs^−1^. For the PPy materials, Jiang et al. employed CNT as a reinforcing component to lessen swelling and shrinkage throughout the doping and dedoping procedures. This CNT/PPy composite has a specific capacitance of 154.5 F∙g^−1^, compared to only 25 F∙g^−1^ for a single PPy, showing a six-fold difference between the two [14]. Canobre et al. also investigated CNT/PPy composites, which had a coulombic efficiency of 99.2% and a specific capacitance of 530 F∙g^−1^ [15,24]. In order to manufacture the PPy/CNT composite on ceramic fabrics, Lee et al. used chemical vapor deposition (CVD) to grow the CNT on the fabrics and then chemical polymerization to coat the surface of the CNT with PPy. This PPy/CNT/ceramic fabric demonstrated a specific capacitance of 152.7 F∙g^−1^ at 1 mA/cm^−2^ and showed exceptional stability even after 5000 cycles) [16,25]. However, there have been no reports of the use of bacteria with PPy/CNT complexes as supercapacitor electrodes.

The creation of a nanostructural composite electrode for use in supercapacitor applications was the study’s main goal. As a bio-template for this, we created active electrodes made of PPy, CNT, and *E. coli* (gram-negative bacteria). Here, it was intended to increase the porosity and surface area of the composite electrode because these microstructures are directly related to the electrochemical performance of composite electrodes. It is well known that the heteroatom nitrogen, which is a component of *E. coli* bacteria, contributes to the increase in capacitance when the bacteria are present. In a study by Kwang et al., it was found that the bacterium *E. coli* improved the electrochemical performance of electrodes in the composite [1]. As a result, when compared to CNT and PPy/CNT composite electrodes, the PPy/CNT–*E. coli* composite electrodes showed better electrochemical performance.

## 2. Materials and Methods

### 2.1. Oxidation of Multiwalled Carbon Nanotubes (CNT)

One gram of CNT (Hanwha Nanotech Co., Korea) was oxidized in 100 mL nitric acid/deionized water aqueous solution in a ratio of 3:1 (wt. %). The dispersion was stirred at 80 °C for 3 h. The oxidized CNT underwent filtration and washing using deionized water until neutrality (pH of 7) was attained. An oven was used to dry the acid-treated CNT at 60 °C for 24 h.

### 2.2. Chemical Synthesis of PPy/CNT

First, 2.7 g of FeCl_3_ (98%, Sigma-Aldrich) was prepared into a 25 mL solution and added to functionalized CNT (0.1 g) uniformly dispersed in 200 mL of deionized water through sonication for 30 min. The resultant solution was stirred for 30 min. Then, 1.4 mL pyrrole monomer (Sigma-Aldrich) was added and stirred for 24 h. After stirring, a precipitate was obtained and filtered. The residue was rinsed thoroughly using deionized water and ethanol. It was then dried overnight at 60 °C under vacuum and the resultant product was a PPy/CNT composite powder.

### 2.3. Preparation of Bacteria

In a 15 mL tube filled with Luria Bertani (LB) broth (LPS solution Inc. Korea), *E. coli* O157:H7 (ATCC) cells were generated and cultured for an entire night at 37 °C while being shaken (150 rpm). By measuring UV absorbance at 600 nm, and optical density 1 (equivalent to 10^9^ cells/mL), the cell density of bacteria was determined [1].

### 2.4. Chemical Synthesis of PPy/CNT–E. coli Composite

FeCl_3_ (2.7 g) and *E. coli* (14 mL) bacteria were prepared into a 25 mL solution and stirred for 30 min. Functionalized CNT (0.1 g) was uniformly dispersed in deionized water by sonication for 30 min. The FeCl_3_/*E. coli* solution was mixed with the sonicated CNT and stirring of the mixture was done for 30 min. The solution was kept under stirring for 24 h after the addition of pyrrole monomer (1.4 mL). The as-prepared powder was filtered with a poly(vinylidene fluoride) (PVdF, SciLab Korea, pore size: 0.45 μm) filter and washed with deionized water and ethanol.

### 2.5. Electrodes Preparation

The working electrodes were used to evaluate the electrochemical properties of the acid-treated CNT. A homogeneous paste consisting of PVDF binder, active materials (CNT, PPy/CNT, and PPy/CNT- *E. coli* series), N-methyl-2-pyrrolidone (NMP, Sigma Aldrich), and vapor-grown carbon fiber was mixed in the weight ratio of 80:15:5. The resultant paste was spread evenly on nickel foam (1 × 1 cm^2^). The nickel foam was dried in a vacuum oven at 60 °C overnight. The active-material mass was 3 mg in the working electrode.

### 2.6. Characterization

Scanning electron microscope (SEM) images of the as-prepared sample were acquired through the use of Hitachi 600. Fourier transform infrared (FT-IR) spectra for the samples (CNT, PPy/CNT, and PPy/CNT–*E. coli* series) with the KBr pellet method were recorded in the frequency range 500–3000 cm^−1^ on a BRUKER-EQUINOX-55 spectrophotometer. An M18Xce X-ray diffractometer accompanied by a Cu Kα radiation was used to take X-ray diffraction (XRD) patterns. The XRD data were taken in the 2𝜃 range of 5–90°. An LEO 1430VP SEM was used as the energy dispersive X-ray analyzer in the energy dispersive spectroscopy (EDS) analysis. The Brunauer–Emmett–Teller (BET) instrument was the micrometric TriStar II 3020 2.00. The slurry (the working electrode) was composed of 5 wt. % PVDF, 85 wt. % active material (CNT, PPy/CNT, and PPy/CNT–*E. coli* series) and 10 wt. % vapor grown carbon fibers (VGCFs). All the calculations were based on the total mass of the electrode. The cyclic voltammetry (CV) of half-cells was measured using the Autolab AUT72432 device (Korea). It was performed from −0.8 to 0.3 V voltage range. A three-electrode system in a beaker was prepared singularly for each CNT, PPy/CNT, and PPy/CNT–*E. coli* on nickel foam as the working electrode with a mass loading of 3 mg/cm^2^, reference electrode (Ag/AgCl), and counter electrode (platinum plate, 1 × 1 cm^2^). All the experimental work for electrochemical characterization was carried out using an electrolyte solution (1 M Na_2_SO_4_ in deionized water).

The specific charge and discharge capacitance of the composites and constituting materials were obtained, and the following equations were used to derive the specific capacitance (SC), measured in Faradays per gram or per area, Galvanostatic cycling profile (CPs), respectively.
(1)$$SCcp=\frac{Q}{\Delta E\times M}$$
(2)$$SCcp=\frac{l\times \Delta t}{\Delta E\times M}$$
where *Q* (in C or mA·s) represents the integrated cathodic (or anodic) charge, ∆E (in V) represents the potential window, *l* (in mA) represents the amount of current applied to the electrode during charging-discharging, ∆t (in sec) represents the amount of time elapsed during the discharge, and M (in mg) represents the loading mass of the active materials.

## 3. Results and Discussion

The chemical species in the electrode materials, CNT, PPy/CNT, and PPy/CNT–*E. coli* composite, are represented by FT-IR spectra in Figure 1. The stretching vibrations of the pyrrole rings, which are associated with two observed peaks at 1537 and 1454 cm^−1^, are symmetric and asymmetric, respectively. At roughly 1301 cm^−1^, the C-N stretching in the pyrrole rings is visible. While the peaks at 991 cm and 920 cm^−1^ are attributed to C-H in-plane and C-H out-of-plane deformation, respectively, the peak at 1160 cm^−1^ reflects the C-H stretching vibration of pyrrole rings. The peaks at 1402 (1454) and 964 (960) cm^−1^ are attributable to the symmetric stretching vibration of phosphoryl groups and the C-O of carboxylic groups, respectively. Furthermore, the P=O vibration in phosphodiesters of nucleic acids, phosphorylated proteins, and polyphosphate products is connected to the bands at 1225 (1238) and 1085 cm^−1^

Figure 2 displays the XRD patterns for the following samples: pure CNT (c), PPy/CNT nano-composites (a), and PPy/CNT–*E. coli* nano-composites (b). Figure 2c demonstrates that the pure CNT’s two XRD peaks were found at 25.78° and 42.84°, respectively, which translate into d-spacings of 0.34 and 0.21 nm. Here, the former is associated with the crystallographic plane (002), while the latter is associated with the plane (100). Notably, Braggs’ law is λ = 2*d* sin 𝜃, where *d* is the interplanar distance, 𝜃 is the diffraction angle, and (λ) is the x-ray wavelength, respectively. On the other hand, it is known that PPy has a large peak at 2 = 20.5° [26], indicating that this polymer is almost amorphous and has disordered π-π stacking. The (002) peak at 2 = 25.78° is, in fact, significantly widened by the presence of PPy, as seen in Figure 2b, and the same is true for the PPy/CNT–*E. coli* sample in Figure 2a. Here, it is noteworthy that although the presence of both Ppy and CNT was justified through XRD, it is not possible to identify the presence of *E. coli* bacteria because these bacteria are amorphous materials.

This section may be divided by subheadings. It should provide a concise and precise description of the experimental results, their interpretation, as well as the experimental conclusions that can be drawn.

The nitrogen-adsorption isotherm was examined using a BET analyzer to look at the porosity of nano-microstructural PPy/CNT and PPy/CNT–*E. coli* samples (Table 1). Because it can both improve capacitance and make electrolyte ions more mobile, a porous structure with large surface areas is crucial for the supercapacitor electrodes in this case. The pore volume of PPy/CNT–*E. coli* is 0.34 cm^3^g^−1^, PPy/CNT has 0.28 cm^3^g^−1^, and pure CNT has 0.16 cm^3^g^−1^, indicating the advantage of the full composite material (PPy/CNT–*E. coli*). Here, the inclusion of PPy ought to improve the redox-active sites, allowing for more effective electron transport. The energy density of energy storage devices is directly influenced by the kinetics and electrochemical activity of electrode materials. Therefore, for good electrochemical performances, the diffusion of charged ions across the interface between the electrolyte and electrode should be sufficiently smooth. The PPy/CNT- *E. coli* has a surface area of 85.93 m^2^g^−1^, which enables electro-active species to participate in the Faradaic redox processes. The meso/macro porous structure could be seen in the morphology of the PPy/CNT–*E. coli* composite. It is known that this type of microstructure is related to the open space between the nearby PPy/CNT–*E. coli* superstructures [27].

When pyrrole monomers were mixed with CNT dispersion, the monomers adhered easily to the CNT surface and then chemically polymerized. The attractive interactions between CNT and pyrrole molecules, as demonstrated by the SEM images in Figure 3, may be the cause of this simple adhesion. According to Figure 3a,b, the diameter of the pure CNT fibers is around 25 nm. However, as shown in Figure 3c,d, the diameter of the CNT fibers expanded up to around 50 nm after pyrrole was polymerized on their surface, providing unmistakable proof that PPy had grown on the surface of the CNT. This PPy coverage on the CNT seems to be all over the CNT fibers/particles, and eventually, PPy encapsulated them. Interestingly, the *E. coli* bacteria template has a tendency to create irregular bumps on the fibril morphologies of the PPy/CNT–*E. coli* composite electrode as shown in Figure 3e,f [12]. Both intracellular and extracellular membrane features in *E. coli* cells are known to have an impact on the final composite electrode. [28]. When the PPy/CNT–*E. coli* composite electrode was made, these *E. coli* qualities led to a wide surface area with rich porosity, which was better than the outcome of the PPy/CNT composite alone. We anticipate that these characteristics, along with structural regularity, should result in the expected capacitance output from the PPy/CNT–*E. coli* electrodes. These structural characteristics encourage charged ion migration (diffusion and drift) into the majority of accessible areas of the composite electrodes.

Cyclic voltammetry was used to assess the CNT, PPy/CNT, and PPy/CNT–*E. coli* at varied scan rates of 5–200 m∙Vs^−1^. The conducing PPy coating on the surface of CNT fibers allows a quick diffusion of charged ions, enhancing the performance of the supercapacitor incorporating this composite electrode [29]. PPy content can be associated with pseudo capacitance contribution because the PPy is able to promote rapid pseudo-Faradaic reactions. The reversible doping/undoping process of anions (D^−^ = I^−^, SO_4_^2-−^) in PPy is inferred to be the key driver of the charge storage of PPy.
$$PPy^{0}+D^{-}↔\left[PPy^{+}\right]D^{-}+e^{-}$$

Comparing the CV voltammograms obtained for CNT and PPy/CNT composite, the PPy/CNT–*E. coli* composite showed the highest total charge. These voltammograms show that the pseudo-capacitance is mostly responsible for the electrochemical capacitance of the PPy/CNT- *E. coli*. The redox current increases with increasing scan rate, which suggests good reversibility of the fast charge-discharge response. The bacteria *E. coli* acts positively as an additional template material to completely use the electroactivity of PPy/CNT. Indeed, the addition of the bacteria in the composite increased the surface area (85.93 m^2^∙g^–1^) of the PPy/CNT–*E. coli* composite when compared with PPy/CNT and CNT. Figure 4 demonstrates that the capacitance of the PPy/CNT- *E. coli* composite electrode originates from both the pseudocapacitance from PPy fibers and the EDLC capacitive behavior from the CNT-*E. coli*-based composite electrode [28]. This finding shows how much better the PPy/CNT–*E. coli* composite electrode performs than either PPy/CNT or CNT materials. The heteroatom moiety from the bacteria is introduced due to the presence of *E. coli*. The presence of the E-coli introduces the heteroatom moiety from the bacteria. The heteroatoms functionalities have a faradaic nature, which induces pseudocapacitance. On the other hand, the small specific capacitance (37 F∙g^–1^) of CNT indicates that the morphology of this CNT electrode is not much optimized, i.e., a very limited surface area and pores.

Figure 5 shows the galvanostatic charge/discharge curves for the CNT, PPy/CNT, and PPy/CNT–*E. coli* systems. Galvanostatic charge/discharge curves were used to calculate the electrodes’ precise capacitance values (Table 2). A high specific capacitance of 173 F∙g^−1^ at 0.2 Ag^−1^ was displayed by PPy/CNT–*E. coli* (Figure 5d). The specific capacitance of the PPy/CNT and CNT materials was 150 F∙g^−1^ and 37 F∙g^−1^, respectively, at the same current density (0.2 Ag^−1^). The symmetrical properties of the charge/discharge curves show that the composite electrodes made of PPy/CNT–*E. coli* have good capacitive performance with reversible redox processes [17].

The cycle test for the PPy/CNT–*E. coli* electrodes is depicted in Figure 6. The specific capacitance retention below 0.2 Ag^−1^ was excellent for the PPy/CNT–*E. coli* electrodes. For instance, the PPy/CNT–*E. coli* electrodes showed a nearly constant specific capacitance even after 1000 cycles, demonstrating the device is electrochemically stable. Importantly, 1000 charge/discharge cycles showed that PPy/CNT–*E. coli* electrodes maintained a 95.95% capacity, demonstrating long-term cycling stability. According to this stability, the PPy fraction of a composite electrode’s volume change did not impact its electrochemical characteristics, resulting in good capacity retention [30].

## 4. Conclusions

In this study, the chemical oxidative polymerization approach was used to create the PPy/CNT–*E. coli* composite. The resulting PPy/CNT–*E. coli* superstructures had a hierarchical structure and were made of phase-pure porous PPy/CNT with a flower-like morphology. In addition to having an astonishingly huge BET surface area (85 m^2^∙g^−1^), they also have meso-/macroporous characteristics brought on by the open space between the nearby individual PPy/CNT–*E. coli* superstructures. These were combined with the mesoporosity present in the nanotubes composed of small nanoparticles. Based on this, the as-prepared materials were used as electrode materials for a supercapacitor. The PPy/CNT–*E. coli* redox reactions were made possible by the numerous electro-active sites that these favorable features supplied. These characteristics ensured a sufficient electrochemical utilization of the proposed composite electrode. PPy/CNT–*E. coli* composite exhibited higher specific capacitance than some of the previously reported PPy/CNT composites at the same current density, as well as large power density, and good stability after 1000 charge/discharge cycles. An important study that successfully demonstrates how PPy/CNT–*E. coli* materials can be used as active materials for supercapacitor electrodes is the measurement of the electrochemical characteristics of PPy/CNT–*E. coli*. The electrochemical activity and electrode kinetics have a significant impact on a supercapacitor’s performance. Therefore, it is essential to improve the kinetics of ion and electron transport in the electrodes and at the electrode/electrolyte interface in order to increase the energy density in supercapacitors at high rates. For the Faradaic redox reactions, it is essential to also engage enough electro-active species that are exposed on the surface. The *E. coli* bacteria was added to the PPy/CNT composite to improve the overall electrode performance. In comparison to the CNT and the PPy/CNT, the specific capacitance of the PPy/CNT–*E. coli* composite electrode was 173 F∙g^−1^ at 0.2 A∙g^−1^. The PPy/CNT–*E. coli* electrode was fully exploited in terms of specific capacitance, as shown by the acquired specific capacitance and the aforementioned features, making it superior to the other samples. The nitrogen species in the *E. coli* bacteria, XPS data (Appendix A) displayed quaternary N(401.1 eV), pyridinic N(398.1 eV), and pyrrolic N(399.4 eV). This gives evidence of the heteroatom contribution from the *E-coli* bacteria incorporated in the PPY/CNT-*E coli* composite. The foreign atoms in bacteria play a superior role in the electrochemical performance of the supercapacitor electrode through faradaic reactions. Since the use of PPy/CNT–*E. coli* composite as supercapacitor electrodes has not yet been published, our preliminary study requires additional research in order to be applied in the future to energy storage devices, such as lithium-ion batteries and supercapacitors.

## Figures and Tables

**Figure 1 materials-15-05804-f001:**
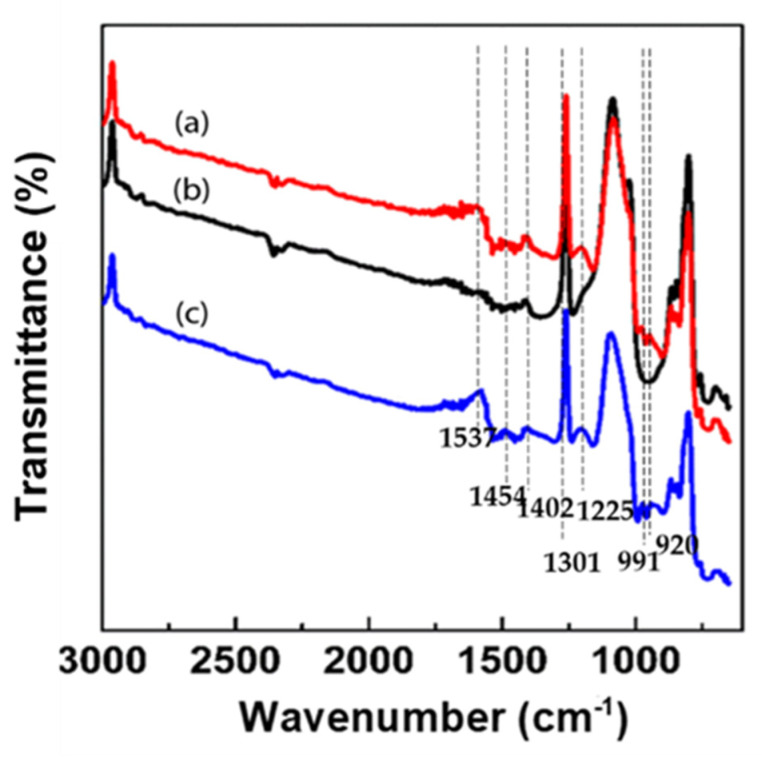
FT–IR spectra of (**a**) PPy/CNT–*E. coli*, (**b**) CNT, and (**c**) PPy/CNT.

**Figure 2 materials-15-05804-f002:**
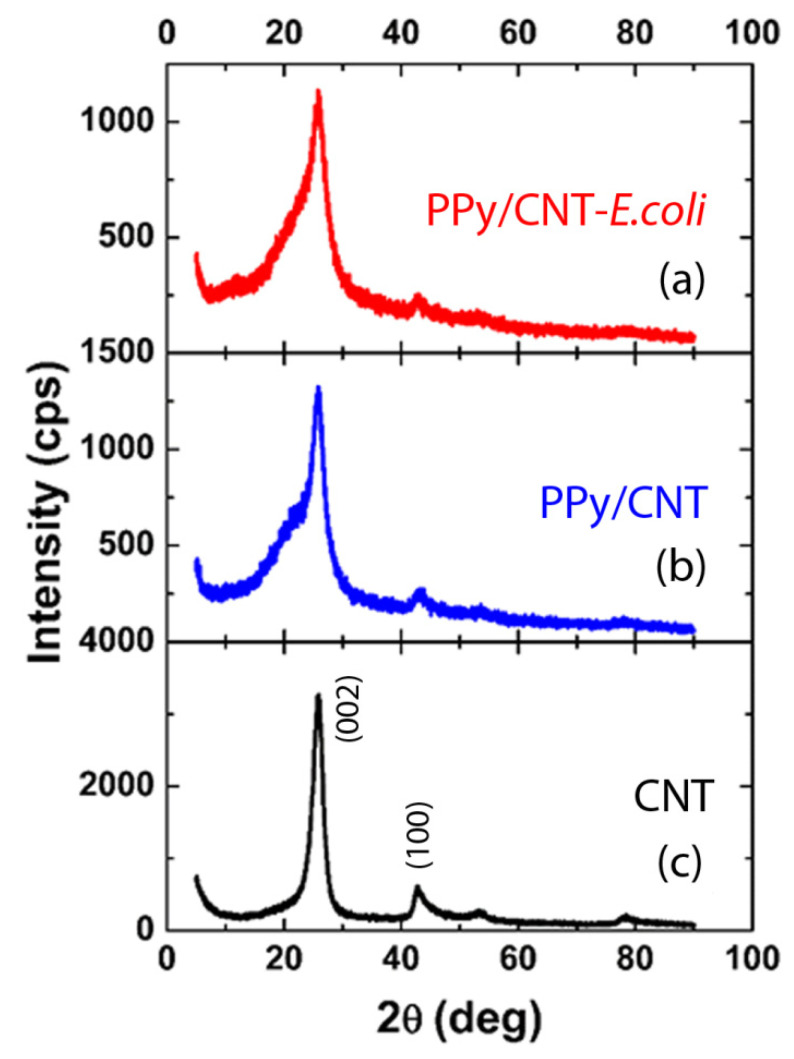
X-ray diffraction pattern of (**a**) Ppy/CNT- *E. coli* nano-composites, (**b**) Ppy/CNT nano-composite, and (**c**) pure CNT.

**Figure 3 materials-15-05804-f003:**
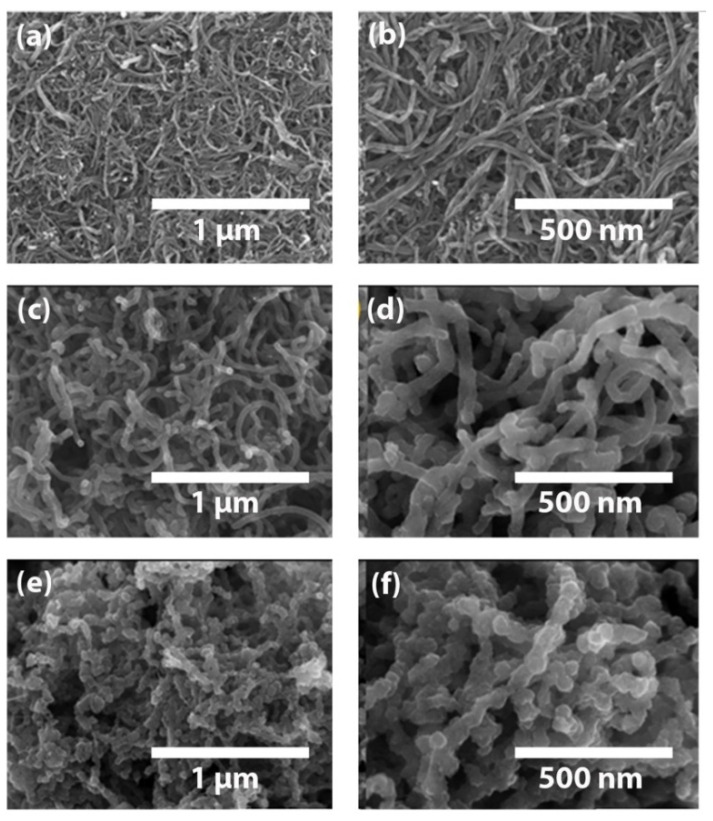
SEM image of (**a**) pure CNT, (**c**) PPy/CNT nanocomposite, (**e**) PPy/CNT- *E. coli* composite (scale bar: 1 μm) and high resolution of (**b**) pure CNT, (**d**) PPy/CNT nanocomposite, (**f**) PPy/CNT- *E. coli* composite (scale bar: 500 nm).

**Figure 4 materials-15-05804-f004:**
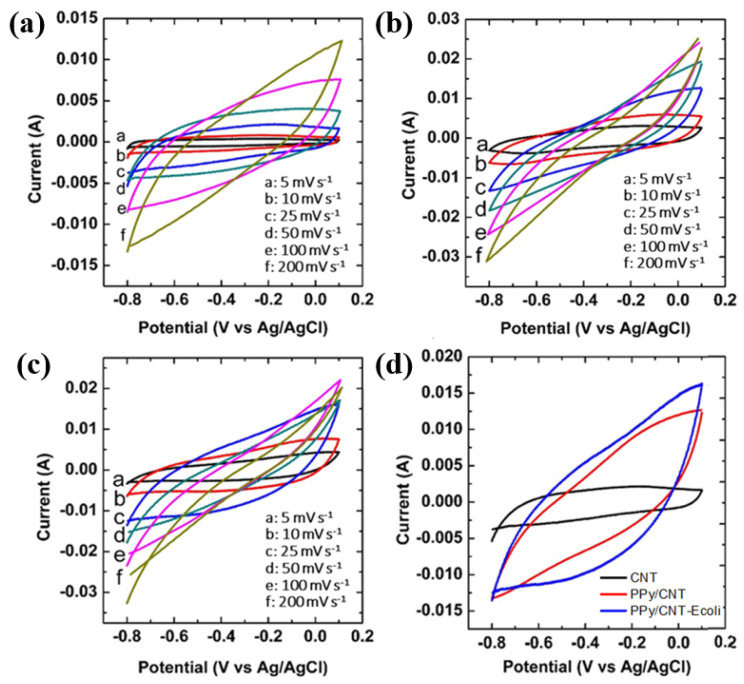
CV curves of (**a**) CNT, (**b**) PPy/CNT composite, (**c**) PPy/CNT*–E. coli* composite, and (**d**) CNT, PPy/CNT, PPy/CNT*–E. coli* at the scan rate of 25 mV·s^–1^.

**Figure 5 materials-15-05804-f005:**
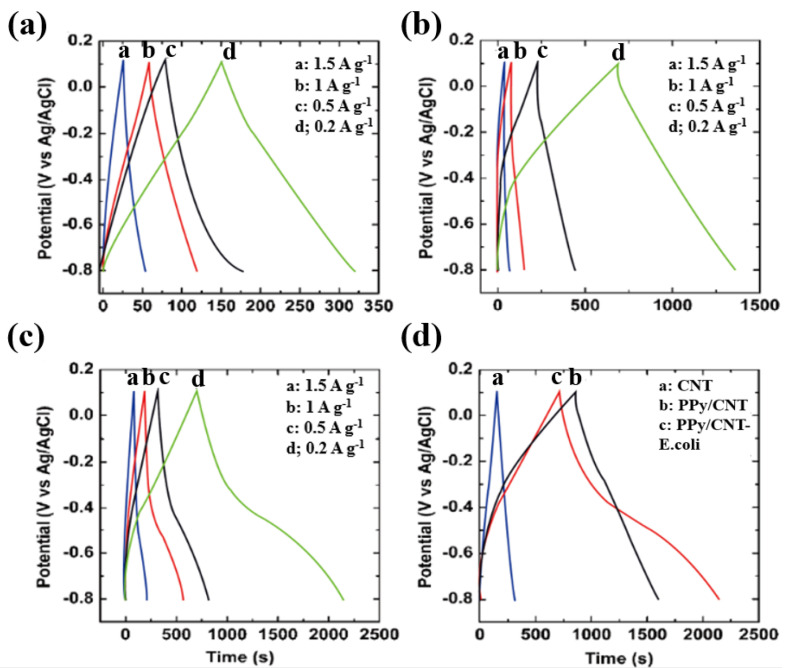
Galvanostatic charge/discharge curves of (**a**) CNT, (**b**) PPy/CNT, (**c**) PPy/CNT–*E. coli*, and (**d**) CNT, PPy/CNT and PPy/CNT–*E. coli* for comparison at 0.2 A·g^–1^.

**Figure 6 materials-15-05804-f006:**
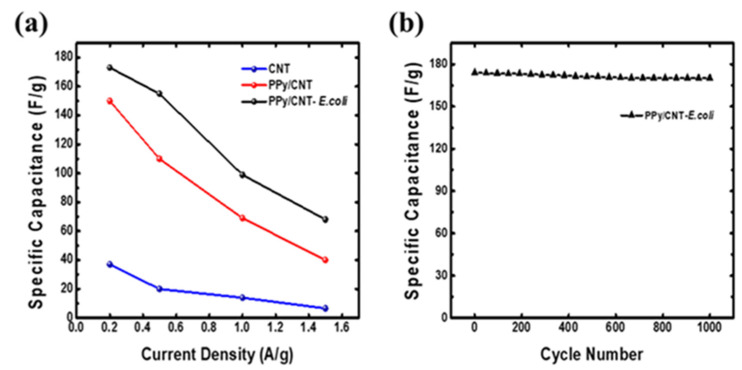
(**a**) Specific capacitances of CNT, PPy/CNT and PPy/CNT–*E. coli* vs. current density and (**b**) PPy/CNT–*E. coli*’s specific capacitance as a function of cycle number.

**Table 1 materials-15-05804-t001:** Surface area, pore-volume, and pore diameter values of CNT, PPy/CNT, and PPy/CNT–*E. coli* composite.

Sample	Surface Area (m^2^g^−^^1^)	Pore Volume (cm^3^g^−^^1^)	Pore Width or Diameter (nm)
CNT	31.60	0.16	20.60
PPy/CNT	80.62	0.28	14.57
PPy/CNT–*E. coli*	85.93	0.34	15.85

**Table 2 materials-15-05804-t002:** Specific capacitance values of CNT, PPy/CNT, and PPy/CNT–*E. coli* calculated from Galvanostatic charge/discharge (GCD) curves.

Sample	Specific Capacitance	Current Density
CNT	37 F∙g^−1^	0.2 A/g
PPy/CNT	150 F∙g^−1^	0.2 Ag
PPy/CNT–*E. coli*	173 F∙g^−1^	0.2 A/g

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
