# Peer review of "Two-Dimensional Heterostructure of PPy/CNT–E. coli for High-Performance Supercapacitor Electrodes"

_materials, 2022, doi:10.3390/ma15175804_

Round 1
Reviewer 1 Report
CNT, PPy and E.Coli are formed into composites used in electrochemical supercapacitors. Additional experimental evidence of the role of E-Coli in the doping/dedoping ion transport activity of PPy is required. Electrochemical quartz crystal microbalance measurements should be performed to provide evidence of ion transport during charge and discharge of the composites, particularly the PPy/CNT-E.coli. In the current form, the postulated enhancing effect of E.coli in providing access of ions to and from PPy is only partially supported.Line 42: "polypyrrole (PPy) has [...] huge capacity [8-12]. However, this kind of organic polymer has some drawbacks (e.g., small conductivity/capacitance" these sentences are contradictory in the capacity/capacitance "huge" or "small"? Please clarify.
Line 46: CNTs are not ceramic materials, why call them "ceramic fillers"? Line 75: "presence of E. coli bacteria is known to enhance the capacitance through the contribution from the heteroatoms", is E. coli used to introduce heteroatoms? Please explain better. Line 83: What CNTs were used? Multi, double, single-walled? Please specify. Also, provided the amount of nitric acid/H2O aqueous solution used per 1 g of CNT. Line 108: "active materials" are these PPy/CNT and PPy/CNT-E.coli? Please specify exactly do not use generic terminology. The same applies to "active material" on Line 124. Line 112: " For the other samples (PPy/CNT and PPy/CNT-E.coli), the same procedure was carried out." what other samples? This is confusing since the "active materials" appeared to be already PPy/CNT and PPy/CNT-E.coli. Please clarify. Line 118: Please provide information on sample preparation and measurement for FT-IR. Was ATR used or KBR pellets or else? Line 126: "slurry was spread out on a graphite electrode" how much slurry per square centimetre? Please specify. Working electrode: It is not clear if the working electrode is (i) the slurry (Line 123) on graphite electrode, or (ii) CNT, PPy/CNT, and PPy/CNT-E.coli on nickel foam. This is confusing, please clarify. Line 135: "FT-IR spectra for the structural analysis" please revise since FT-IR does not provide information on the structure of materials, but on the chemical groups/species present in materials. Line 141: "1544 cm-1 for N–H vibration [...] originated from the E.coli bacterial" please revise since there is no clearly distinguishable IR peak for PPy/CNT-E.coli compared to CNT and PPy/CNT. Line 145: What is a "versatile" vibration? Figure 1: Please specify that the "Intensity (a.u)" is in the plot, transmittance? Line 156: "Braggs’ law is λ= 2d sine?" should be " Bragg's law is λ= 2d sin?" please correct. Line 161: "However, it is very interesting to see that the shoulder peak at 2? = 21.18° (Figure 2b) became very weak when the composite was PPy/CNT-E.coli in Figure 2a, indicating the change of microstructural morphology in the samples" the change of intensity of the shoulder at 21° is not that evident. Line 228: "The conducing PPy coating on the surface of CNT fibers might allow a quick diffusion of charged ions, enhancing the performance of the supercapacitor incorporating this composite electrode", please include a discussion of the Faradaic redox activity of PPy and review this sentence since it is not obvious that the current is capacitative. PPy is electroactive and current flow is recorded in the process redox process PPy+/PPy0 (doping/dedoping process). Figure 4d: In comparing the CVs of PPY/CNT, PPY/CNT-E.coli what is the effective contribution of E.Coli to the redox activity of PPy? What evidence is there of the electrolyte ions accessibility in the entire active sites of the composite materials? English editing is necessary.Author Response
Please see the attachment.

Reviewer 2 Report
The paper presents a characterisation of PPy/CNT-E.coli using different methods (FTIR, X-ray diffraction, BET , SEM) pointing the influence of e-coli on the microstructural morphology of the composite. Yet, the electric properties characterisation of the sample is superficial and the conclusions are not sustained by the data. For instance, the authors present charge/discharge curves but they are not properly described, and there are some results in lines 255-259 that should be pointed in Fig.5 but the connection of the results with the figure is not given (where does the value of 173 F g–1 at 0.2 A·g–1 appears in Fig.5 c?). The specific capacitance versus the current density and versus the cycle number are only presented for the PPy/CNT-E.coli without any data to compare with the other studied componds (CNT, PPY/CNT, PPY/CNT-E.coli). Thus the conclusions are not sustained by the data so the manuscript needs a major revision before being considered for publication.
Reviewer 3 Report
The paper is interesting and well prepared, however, before publication I suggest adding some more information according to my suggestion:
In the article, in the introduction, there is little information about the methods of producing supercapacitors, please extend the literature studies on this subject, with particular emphasis on the literature from the last two years
in section 2.3 the authors describe that they use e-coli bacteria. There is no information on what stage of development of these bacteria. (Information about the stages of development, e.g. https://doi.org/10.3390/s21010183)
what happened with the bacteria after electrodes heating at 60C?
Reviewer 4 Report
The manuscript “Two-Dimensional Heterostructure of PPy/CNT-E.coli for High-Performance Supercapacitor Electrodes” addresses an interesting topic and clearly describes the results.
Moreover, the paper also uses a good English. Therefore, it may be recommended for publication after minor revision:
1) Authors should insert the value of the peaks in the FTIR analysis (fig. 1)
2) Authors should describe in detail all results, in particular: insert statistical analysis
3) In the introduction part it is recommended to add some more discussion. The following publication is recommended to fulfill this section:
CATAURO, Michelina, et al. Antibacterial properties of sol–gel biomaterials with different percentages of PEG or PCL. In: Macromolecular Symposia. 2020. p. 1900056
Round 2
Reviewer 1 Report
The authors have not provided additional direct experimental evidence of the role of E.coli in the doping/dedoping ion transport activity of PPy.
BET is a measure of gas adsorption not of ion transport. The authors make the assumption that larger pore volume delivers greater ion transport, this is an assumption not experimental evidence.
Since the authors have no access to EQCM, they are welcome to suggest and perform measurements with another technique that measure ion transport, not gas sorption. Without direct experimental evidence of the effect of E.Coli on ion transport, the parts of the manuscript referring to it should be either removed or restated as based on assumption.
Reviewer 2 Report
The authors performed the required revision.
I believe that the work can be published in the form presented.
Round 3
Reviewer 1 Report
Although direct evidence of improved ion transport is still missing, the reviewer appreciates the additional effort put in providing further support on the role of E-coli on improve ion transport in composite materials.
